# Fast Eccentric Movement Tempo Elicits Higher Physiological Responses than Medium Eccentric Tempo in Ice-Hockey Players

**DOI:** 10.3390/ijerph18147694

**Published:** 2021-07-20

**Authors:** Mariola Gepfert, Robert Trybulski, Petr Stastny, Michał Wilk

**Affiliations:** 1Institute of Sport Sciences, Jerzy Kukuczka Academy of Physical Education in Katowice, 40-065 Katowice, Poland; stastny@ftvs.cuni.cz (P.S.); m.wilk@awf.katowice.pl (M.W.); 2Department of Medical Sciences, The Wojciech Korfanty School of Economics, 40-065 Katowice, Poland; rtrybulski@o2.pl; 3Provita Zory Medical Center, 44-240 Zory, Poland; 4Department of Sport Games, Faculty of Physical Education and Sport, Charles University, 162 52 Prague, Czech Republic

**Keywords:** cortisol, testosterone, insulin-like growth factor 1 (IGF-1), growth hormone (hGH), resistance training, conditioning

## Abstract

Background: Resistance training is a significant part of ice-hockey players’ conditioning, where optimal loading should ensure strength development and proper recovery. Therefore, this study aimed to compare the acute physiological responses to fast and medium movement tempo resistance exercises in ice-hockey players. Methods: Fourteen ice-hockey players (26.2 ± 4.2 years; 86.4 ± 10.2 kg; squat one repetition maximum (1RM) = 130.5 ± 18.5) performed five sets of the barbell squat and barbell bench press at 80% 1RM until failure in a crossover design one week apart using either 2/0/2/0 or 6/0/2/0 (eccentric/isometric/concentric/isometric) tempo of movement. The blood samples to evaluate the concentration of cortisol, testosterone, insulin-like growth factor 1 (IGF-1), and growth hormone (hGH) were taken before exercise, 3 min after the last set of the squat exercise, 3 min after the last set of the bench press exercise, and after 30 min of recovery. Results: The 2/0/2/0 tempo resulted in a higher number of repetitions (*p* < 0.001) and lower time under tension (*p* < 0.001) in the squat and bench press exercises compared to the 6/0/2/0 movement tempo. The endocrine responses to exercise were significantly higher during the 2/0/2/0 compared to the 6/0/2/0 movement tempo protocol for IGF-1, hGH, and cortisol (*p* < 0.01). There were no differences in testosterone responses between exercises performed with fast and medium movement tempos. Conclusion: Fast eccentric tempo induced higher cortisol, IGF-1, and hGH responses compared to the medium tempo. Therefore, fast eccentric movement tempo seems to be more useful in eliciting training stimulus than medium eccentric tempo during resistance training in ice-hockey players. However, future studies are needed to confirm our findings.

## 1. Introduction

Resistance exercises are a crucial part of ice-hockey players’ conditioning, where optimal loading should ensure strength development and recovery by increasing maximal strength and power output [1,2,3]. Furthermore, the level of strength and power output is an important predisposition to succeed at the elite NHL performance level [4]. The current knowledge on ice-hockey resistance training programs is apparent in regard to exercise intensity, load, rest intervals, and exercise selection [2,3,5] without detailed description of the optimal tempo of movement during such exercises [6]. Movement tempo in resistance exercise is usually described using a sequence of digits (e.g., 2/0/3/0), where each digit defines the duration of a particular phase of the movement using a four-digit combination: eccentric, isometric, concentric, and isometric [7,8,9]. According to the movement tempo classification, fast tempo is the duration of a single repetition between 2 and 5 s, medium movement tempo occurs when the duration of a single repetition is between 5 and 10 s, and slow movement tempo takes place when the duration of a single repetition is above 10 s [8].

Ice-hockey players perform 22.3 ± 4.9 shifts per game lasting about 45 s, where 23.0 ± 12.6% is covered by gliding, 42 ± 7.1% by slow skating [10], and 17.6 ± 6.0% by high-intensity skating, which requires high levels of anaerobic power 11.6 ± 1.3 (W/kg) [11]. The resistance training performed by ice-hockey players should maintain or increase non-specific conditioning levels, improve specific fitness, and ensure the physiological responses for proper recovery. Thus resistance training programs for hockey should elicit loading that stimulates optimal physiological and hormonal responses. The metabolites and hormonal accumulation during and after resistance training are the primary stimuli for changes in strength and muscle hypertrophy, where the key hormones in training adaptations are testosterone (T), growth hormone (GH), cortisol (C), and insulin-like growth factor 1 (IGF-1) [8]. Taking into account the post-exercise hormonal responses, optimal training loads should have a limited post-exercise response in cortisol levels, and at the same time an induced high post-exercise response of hGH, T, and IGF-1 [12]. It seems that these hormones and growth factors are influenced by resistance exercise movement tempo [8,12,13,14]. Previous research has shown that different movement tempos impact acute physiological responses, including hormone and blood lactate concentration [8]. It has been suggested that a medium movement tempo could increase the metabolic [15,16] and hormonal response provided by resistance exercise [8,14,17]. Medium movement tempo increases time under tension (TUT) during a set, which increases the volume of effort [13]. Higher volume protocols with medium movement tempo and greater metabolic demands lead to increased hormonal responses compared to faster movement tempo [18,19]. Furthermore, the physiological effect of medium movement tempo during resistance exercise can be similar to what occurs during resistance exercise with blood flow restriction [20]. The effects of exercise regimens with restricted muscular blood flow are likely mediated by the stimulated secretion of growth hormone and by intramuscular accumulation of metabolic byproducts, moderate production of reactive oxygen species promoting tissue growth, and additional recruitment of fast-twitch fibers under hypoxic conditions [20]. However, most studies about acute impact of different movement tempos were performed in recreationally trained subjects, while such assessments have not been performed in elite athletes who would be the most likely candidates for advanced resistance-training methods, such as varied movement tempo.

Off-ice ice-hockey conditioning requires knowledge about the specificity of the desired training conditions [2,3,5,21,22,23,24], where, e.g., the ice-hockey take-off is performed in a more extended period (0.33 s) than movements such as sprint ice skating (0.08–0.25 s), or single leg hops (0.124 s) [23]. Since ice-hockey players spend approximately 39% of the match in a two-foot glide position [24], their on-ice sprint conditions are related to bilateral squat and jump performance [4,25], thus resistance training that includes squats can improve on-ice sprint skating [26]. Upper limb strength in ice-hockey players is typically evaluated by the bench press performance [4,10,11], while the bench press exercise belongs to the main complex exercises used in off-ice resistance training [3]. Although there is a general agreement in selecting squats and bench press for off-ice training, there is a lack of empirical data on the optimal duration of the eccentric phase of the movement. It has been shown that changes in the eccentric movement duration can influence acute hormonal and kinematic responses and may provide significant strength training variability [13]. Therefore, different movement tempo may be an essential factor in resistance training performed by ice-hockey players, who are highly adapted to isometric and controlled eccentric contractions during ice skating [23,27,28].

While the total number of studies related to the impact of different movement tempo is relatively high, there is a lack of research representing particular populations, especially people advanced in resistance training and competitive athletes [9]. Only a small number of studies included highly trained athletes, who would be the most likely candidates for advanced resistance-training methods, such as varied movement tempo resistance training [8,9]. Currently only two studies have considered acute impact of different movement tempos during resistance exercise on post-exercise hormonal responses in subjects with significant resistance training experience. Wilk et al. [13] showed that post-exercise increases of testosterone were greater after the bench press exercise (5 sets, maximal number of performed repetitions, load 70% 1RM) performed with a medium compared to a fast movement tempo (6/0/2/0 vs. 2/0/2/0) in subjects with at least five years of resistance training experience (5.7 ± 1.29 years). In contrast, Wilk et al. [14] did not show changes in post-exercise hormone concentrations (T, hGH, cortisol, IGF-1) after the squat exercise (5 sets, maximal number of performed repetitions, load 80% 1RM) performed with medium vs. fast tempo of movement (5/0/3/0 vs. 2/0/2/0) in a group of powerlifters. Therefore the type of exercise can be a significant factor influencing acute post-exercise hormonal responses following exercise performed with different movement tempos. However, currently there are no studies regarding the impact of movement tempo on acute post-exercise hormonal changes during a research protocol containing simultaneously squat and bench press exercises, two basic exercises for developing strength and power output of the upper and lower limbs. Furthermore, there are also no studies related to the impact of different movement tempo on acute physiological responses with subjects habituated to resistance exercise performed with different movement tempos. Finally, there is a lack of studies regarding the acute effects of movement tempo on physiological responses in elite ice-hockey players.

Therefore, the objectives of the study were to compare the acute physiological responses of fast and medium eccentric movement tempo during resistance exercises in elite ice-hockey players. We hypothesized that the medium eccentric movement tempo would provide longer time under tension and greater acute hormonal responses than fast movement tempo.

## 2. Material and Methods

The experiment was performed in a randomized crossover design, where each participant performed eight familiarization sessions, one session of 1RM testing, and two main testing sessions 7 days apart. During each experimental session, the participants performed 5 sets of barbell squats (SQ) and 5 sets of the bench press exercise (BP) against a load of 80%1RM with different movement tempos: fast (2/0/2/0) or medium movement tempo (6/0/2/0), where the 6/0/2/0 movement tempo denotes a 6 s eccentric phase, no pause during the transition phase (0), a 2 s concentric phase, and 0 refers to no pause between the completion of the concentric phase and the beginning of the next eccentric phase (Figure 1). We decided to use the 2/0/2/0 movement tempo as the fast one based on previous results, which indicate that the ranges of volitional duration of the concentric movement in the squat and the bench press exercise at 80% 1RM were 0.76–1.29 and 0.90–1.93 s, respectively [29]. The medium eccentric tempo (6 s) has been chosen as duration regularly used by research participants in their current resistance training routines. The testing session also included a countermovement jump test (CMJ) before and after the SQ and the BP (2 min rest interval) to assess lower limb fatigue levels based on changes in power output. During the experiment, venous blood samples were collected from the antecubital vein. In each case, 10 mL of venous blood was drawn to determine pre-squat and post-exercise values of the following biochemical variables (cortisol, testosterone, GH, IGF-1). The blood samples were taken before exercise, 3 min after the last set of the squat exercise and the bench press exercise, and after 30 min of recovery. All familiarization and experimental sessions were performed during the off-season, and during the preparatory period to the regular season. The main experimental sessions were performed on the same day of the week (Monday), preceded by two days of rest. The routine training workout was maintained on the other days of the weekly microcycle.

### 2.1. Participants

Fourteen athletes representing a professional first league ice-hockey team, experienced in resistance training (8.2 ± 4.2 years), volunteered for the study after completing an informed consent form (age = 26.2 ± 4.2 years; body mass = 86.4 ± 10.2 kg; SQ 1RM = 130.5 ± 18.5; BP 1RM = 100.6 ± 13.0 kg). Subjects were allowed to withdraw from the experiment at any moment and were free from musculoskeletal disorders. The subjects were instructed to maintain their normal dietary habits throughout the study and not to use any supplements or stimulants for the experiment’s duration. Subjects were informed about the study’s benefits and potential risks before providing their written informed consent for participation. The Bioethics Committee approved the study protocol for Scientific Research at the Academy of Physical Education in Katowice, Poland (10/2018) which was performed according to the ethical standards of the Declaration of Helsinki 2013.

### 2.2. Familiarization Session and 1RM Strength Test

Four weeks before the main experiment, the athletes performed familiarization sessions twice per week. During the familiarization sessions, the athletes performed resistance training including the squat and bench press exercises performed with a 2/0/2/0 or 6/0/2/0 tempo of movement. The familiarization sessions were conducted to restrict possible learning effects. One week before the main experiment 1RM testing was performed. The subjects arrived at the laboratory at the same time as the upcoming experimental sessions. They cycled on an ergometer for 5 min, followed by a general upper body warm-up of 10 bodyweight pull-ups and 15 bodyweight push-ups. Next, the subjects began the 1RM test for the SQ and BP exercises. The subjects performed 15, 10, and 5 SQ repetitions using 20%, 40%, and 60% of their estimated 1RM. The first testing load was set to an estimated 80% 1RM and was increased by 2.5 to 10 kg for each subsequent attempt. Then the subjects executed one repetition with a 5 min rest interval between successful trials, and repeated this process until failure. After a 10 min recovery period, the participants completed the 1RM test for the BP with an identical test protocol as for the SQ [13,29].

### 2.3. Experimental Sessions

The subjects performed the squat and the bench press exercise against a load of 80% 1RM either with 2/0/2/0 or 6/0/2/0 tempo of movement. In each exercise, the subjects performed 5 sets with a maximal number of repetitions with a metronome guided movement tempo in the eccentric and concentric phases (Korg MA-30. Korg. Melville. New York, NY, USA). The rest interval between sets was 3 min. The rest interval between exercises lasted 5 min. The interval between both experimental test protocols was 7 days to avoid the accumulation of fatigue. Time under tension and the number of performed repetitions were obtained manually from the recorded data. To ensure manual data collection reliability, four independent persons made the data analysis from the Sony camera. During the experimental sessions, the number of performed repetitions (REP) in particular sets of the squat (SQ-REP), and the bench press exercises (BP-REP) were recorded. The time under tension (TUT) in all 5 sets of the squat (SQ-TUT) and the bench press (BP-TUT) was also registered.

### 2.4. Squat Exercise

The position for the squat was controlled and was identical in every attempt. The athletes started from an upright position, with the knees and hips fully extended, the stance approximately shoulder-width apart with both feet positioned flat on the floor in parallel or externally rotated to a maximum of 15°. The bar rested across the back at the level of the acromion. Stance width and feet position were individually adjusted and carefully replicated on every lift. The bar was required to remain in contact with the back and shoulders at all times. From this position, they were required to descend until making contact the upper leg was horizontal [30,31].

Knee wraps and safety belts were not used. At all times during the testing protocol and warm-up sets, three spotters were present. The squat was performed with an Eleiko IPF barbell (2.9 cm diameter, length 1.92 m).

### 2.5. Bench Press Exercise

During the bench press test protocol hand placement on the barbell was set at 150% of the individual bi-acromial distance. The positioning of the hands was recorded to ensure consistent hand placement during all experimental sessions. The BP was performed with an Eleiko IPF barbell (2.9 cm diameter, length 1.92 m) and on an Eleiko competition bench [32].

### 2.6. Countermovement Jump Test

The participants were instructed to perform the CMJ on a force plate with maximal effort. The CMJ was measured using an AccuPower force plate (AMTI OR6-7-1000; Watertown, MA, USA), allowing ground reaction forces to be assessed with 1.000 Hz sampling to determine jump velocity and power output. The CMJ was performed with both hands on the waist while making a downward movement approximately to 90° knee flexion followed by a maximum effort vertical jump. The investigators also encouraged the athletes verbally for maximum performance to reach peak velocity (V_max_ in m/s) and peak power output (P_max_, W/kg) [33,34]

### 2.7. Biochemical Analysis

During the experiment, venous blood samples were collected from the antecubital vein. In each case, 10 mL of venous blood was drawn to determine pre and post-exercise biochemical values of the analyzed variables (cortisol, testosterone, GH, IGF-1). The post exercise values were taken 3 min after the cessation of the last set of squats, 3 min after the cessation of the last set of the bench press, and after 30 min of recovery. Commercially available radioimmunoassay evaluations were performed for the assessment of testosterone ng/dl (Cobas), GH ng/mL (Immulite 2000 XPi), IGF-1 ng/mL (Immulite 2000 XPi), and cortisol µg/dl (Cobas). Each sample underwent six analyses to ensure accurate results.

### 2.8. Statistical Analyses

All statistical analyses were performed in STATISTICA software 13.3 (Tibco, Palo Alto, CA, USA) at alfa level 0.05. The biochemical analysis reliability was calculated by the intra-class correlation coefficient and data normality by the Shapiro Wilk test. The calculated sample size for non-inferiority and superiority tests in cross-over designs was n = 13 to achieve 80% power for both values at *β* = 0.2. The differences in the number of performed repetitions (repetition × set × tempo), TUT (TUT × set × tempo), fatigue (CMJ × set × tempo), and biochemical markers (biochemical value × set × tempo) were analyzed with the repeated measures ANOVA. The ANOVA *p* < 0.05 and result of Tukey post hoc test was considered significant at effect size determined by partial eta square η^2^ classified according to Larson-Hall [35] and Cohen [36], where η^2^: 0.02, 0.13, 0.26 were considered as small, moderate, and large effects respectively. Cohen *d* was used to express the effect size between each condition due to fatigue and biochemical markers considering d 0.2, 0.5, 0.8, 1.2 as small, medium, large, and very large effect, respectively.

## 3. Results

There were no significant differences in the SQ-REP [n]; BP-REP [n]; SQ-TUT [s]; BP-TUT [s] between the data collected by four evaluators. The data normality was not disrupted, and the descriptive values of jumping fatigue, and biochemical markers are presented as mean values and standard error (Table 1), where baseline (pre-exercise) values did not differ between tempo protocols. The ICC for the biochemical analysis varied from 0.88 to 0.99 for the 6 samples and were considered valid.

The ANOVA analyses for the number of repetitions was different for particular sets (F_12,208_ = 4.1, *p* < 0.001. η^2^ = 0.19) and exercises (F_4,208_ = 149. *p* < 0.001. η^2^ = 0.82). The post hoc tests showed that during the squat exercise the maximal number of performed repetitions was significantly higher in tempo 2/0/2/0 compared to 6/0/2/0 tempo in each set (*p* < 0.01), and during the bench press exercise the maximal number of performed repetitions was significantly higher in the 2/0/2/0 tempo compared to the 6/0/2/0 tempo in each set (*p* < 0.001; Figure 2).

The ANOVA showed that the TUT was different between sets (F_12,208_ = 121, *p* < 0.001. η^2^ = 0.82) and exercises (F_4,208_ = 83. *p* < 0.001. η^2^ = 0.70). The post hoc showed that the SQ-TUT in 6/0/2/0 tempo was significantly higher compared to the SQ-TUT 2/0/2/0 tempo in each set (*p* < 0.01), and the BP-TUT 6/0/2/0 was significantly higher compared to the BP-TUT 2/0/2/0 in each set (*p* < 0.01; Figure 2).

The results of ANOVA for fatigue showed that results of the CMJ performance changed significantly in case of V_max_ (F_2,64_ = 22, *p* < 0.001, η^2^ = 0.41) during the training sessions and were significantly different between exercise protocols (F_2,64_ = 3.5, *p* < 0.034, η^2^ = 0.09). The post hoc tests showed a significant decrease in V_max_ after the squat exercise at both tempo protocols (*p* < 0.01), and the 2/0/2/0 tempo protocol resulted in a higher decrease compared to the 6/0/2/0 movement tempo protocol (Figure 3). After the bench press, the V_max_ in both tempo protocols was significantly lower compared to baseline results (*p* < 0.01), but the V_max_ post-bench press for the 2/0/2/0 tempo was significantly higher compared to post-squat results (*p* < 0.01; Figure 4).

Significant differences were found in the CMJ P_max_ (F_2,64_ = 12, *p* < 0.001, η^2^ = 0.27) during the training sessions and between exercise protocols (F_2,64_ = 3.3, *p* < 0.043, η^2^ = 0.09). The post hoc test showed that P_max_ significantly decreased after the SQ 2/0/2/0 tempo (*p* < 0.01; Figure 3), and the 2/0/2/0 tempo protocol resulted in significantly lower values compared to the 6/0/2/0 tempo protocol. After the BP exercise the P_max_ in both tempo protocols was significantly lower compared to baseline P_max_ (*p* < 0.01) and showed no differences compared to post-squat values (Figure 3, Table 2 and Table 3).

Significant differences between cortisol concentration were found during and after the training sessions (F_3,78_ = 24, *p* < 0.001, η^2^ = 0.48) and between different movement tempo protocols (F_3,78_ = 5.8, *p* < 0.001, η^2^ = 0.18). The post hoc test showed that cortisol concentration significantly increased after the SQ exercise compared to baseline (*p* < 0.01), and increased significantly after the BP exercise compared to all other measures (*p* < 0.01). The concentration of cortisol was also significantly higher 30 min post-exercise compared to baseline values in the 2/0/2/0 tempo protocol (*p* < 0.01). In the 6/0/2/0 tempo protocol, the cortisol concentration after the squat and bench press was significantly higher compared to baseline values and compared to 30 min post-exercise (*p* < 0.01; Figure 4). Besides the baseline, all cortisol values were higher in the 2/0/2/0 tempo compared to the 6/0/2/0 exercise protocol (*p* < 0.01; Figure 4, Table 2 and Table 3).

Significant differences between GH concentrations were found during and after the training sessions (F_3,78_ = 27, *p* < 0.001, η^2^ = 0.51) and different tempo exercise protocols (F_3,78_ = 3.8, *p* < 0.042, η^2^ = 0.10). The post hoc tests showed that GH concentration was significantly higher after the squat compared to all other measures (*p* < 0.01), significantly increased after the bench press exercise compared to baseline and compared to 30 min post-exercise (*p* < 0.01), and increased 30 min into recovery compared to the baseline in both protocols (*p* < 0.01). Besides baseline, all GH values were significantly higher in the 2/0/2/0 exercise protocol compared to the 6/0/2/0 protocol (Figure 4, Table 2 and Table 3).

Significant differences between IGF-1 concentrations were found during and after the training sessions (F_3,78_ = 5.5, *p* = 0.002, η^2^ = 0.17). The post hoc test showed a significant increases of IGF-1 concentration after the squats compared to all other measures in both exercise protocols (*p* < 0.01) and a significant increase after the bench press exercise compared to baseline and compared to 30 min of recovery in the 2/0/2/0 tempo protocol (*p* < 0.01). The IGF-1 concentration after the squat and bench press exercise was significantly higher following the 2/0/2/0 compared to the 6/0/2/0 movement tempo protocol (Figure 4, Table 2 and Table 3).

Significant differences between testosterone concentrations were found during and after exercise (F_3,78_ = 56, *p* < 0.001, η^2^ = 0.68). The post hoc test showed significant increases in testosterone concentration after the squat and bench press exercises compared to pre-squat and 30 min into recovery in both protocols (*p* < 0.01; Figure 4, Table 2 and Table 3).

Particular hormonal responses were different among the athletes, however with similar trends (Figure 5).

## 4. Discussion

The main finding of this study was that ice-hockey players had similar testosterone responses after medium and fast eccentric movement tempo during resistance exercise. However, fast eccentric tempo induced higher cortisol, IGF-1, and growth hormone responses compared to the medium tempo. Therefore, this result justifies the use of fast eccentric tempo during resistance training programs in ice-hockey players to induce a higher post-exercise increase in hormonal responses, which may be an important factor influencing muscular strength and hypertrophy adaptive changes. However, the fast movement tempo exercise protocol elicited greater muscular fatigue, which should be taken into account when programming the recovery time during particular training micro-cycles.

The results of the presented study are contradictive with previous studies showing that a medium movement tempo was more effective in inducing higher acute hormonal responses following resistance exercise [8,13,16], as well as contradictive with studies which do not show differences in acute hormonal responses between different movement tempos [14,37,38]. However, the differences between the results of the presented experiment and other studies can be related to the specificity of the study participants. Ice-hockey players are well adapted to isometric and controlled eccentric loading in the squat position below 90°of knee flexion [10,24,28], therefore increasing the duration of the eccentric part of an resistance exercise does not provide additional stimulus, as reported in other studies [13,16]. The factor of adaptation to isometric lower limb loading is pronounced during the ice-hockey season with every day on-ice training. Additionally, it should be noted that this research procedure assumes conducting a 4-week familiarization session with the exercises performed with different movement tempo to restrict possible learning effects. According to the authors’ knowledge, there were no previous studies related to the impact of different movement tempos on acute hormonal and metabolic responses in participants or athletes habituated to resistance exercise performed with medium eccentric movement tempo, which also could have an impact on the obtained results. The longer movement duration increases TUT during resistance exercise [32], which is usually the main factor inducing higher post-exercise hormonal changes in medium or slow movement tempo. Such increases of TUT could be a stimulating factor especially when medium movement tempo is used as an alternative method of resistance training. However, when such training intervention (with longer maximal TUT) is performed commonly in training routines, it may no longer be sufficient to stimulate acute hormonal responses. Furthermore the Pareja-Blanco et al. [39] suggested that greater force generation would be required to perform faster movements, which would result in greater recruitment of muscle fibers with higher glycolytic potential despite the longer TUT during training with medium movement tempo. It has been speculated that local accumulation of anaerobic energy metabolites, such as lactate, stimulate the secretion of anabolic hormones [40] which may also be the basis for explaining the obtained results. Moreover, as it was observed in our study, the fast exercise movement tempo protocol elicited greater muscular fatigue, which may be explained by higher total concentric work performed during the exercise with fast movement tempo (greater number of performed repetitions). Thus, increasing the TUT as it was observed for the medium movement tempo does not provide more intense acute post-exercise hormonal responses in subjects habituated to medium or slow movement tempo, yet a greater number of repetitions during the concentric phase of the movement does. Our results seem to be contradictive with previous studies showing that a longer eccentric phase increases the hormonal response [8,13], however, there is always the question of whether TUT or the number of performed repetitions are superior in eliciting a greater endocrine response. Since cortisol responses gradually increased after the squat and bench press exercises in both protocols, we can assume that both protocols were sufficiently exhausting, yet the hockey players responded more intensively to the training protocol with a higher number of performed repetitions and a faster movement tempo (2/0/2/0). Although the testosterone and GH were significantly elevated after the squat, their increase after the bench press was lower. This may be explained by the fact that the anabolic response to exercise is partially related to exercise volume and may even decrease along with fatigue. In this case, the bench press provided a lower lifted workload and was performed after a more exhaustive squat exercise. However, a similar response in testosterone and overall hormonal secretion shows that both exercise tempo protocols provide a positive anabolic stimuli, which can be beneficial for ice-hockey players. Therefore, we cannot conclude that medium movement tempo is ineffective, but less effective in inducing endocrine responses than a fast eccentric movement tempo.

According to the results of our study, it can be concluded that resistance exercise sessions targeted at recovery (anabolic responses) should be performed in ice-hockey players with a fast eccentric movement tempo. This is especially relevant if the athletes can maintain proper exercise technique throughout the whole training session. The application of the fast tempo of movement should be performed for non-specific exercises [2,3] which should not overlap with specific ones. On the other hand, medium movement tempo during resistance exercises could be applied if players have problems with exercise technique or during hockey-specific exercises [5]; however, this training will have a smaller effect on GH and IGF-1 responses than fast movement tempo and will provide a smaller anabolic (regenerative) response. In general, training with prolonged eccentric tempo is very specific, and induces most of the adaptive changes through increased time under tension. On the other hand fast tempo of movement during resistance exercises induce their metabolic effect through a higher number of performed repetitions.

The presented training protocol may be regarded as non-specific for ice-hockey players since two main complex exercises were used in their bilateral basics. During the season, a high dominance of specific loading may elicit over-use injuries, and therefore more general exercises should be used to balance the specific ones. Thus, it is essential that exercises like bilateral squats or the bench press provide anabolic responses without over-stressing the same ligaments and muscles used during hockey-specific movements like hip abduction/adduction [28,41,42]. Therefore, the presented resistance exercise protocols should be used for ice-hockey as non-specific loading aimed to provide an anabolic stimuli.

This study’s principal limitation is the cross-sectional evaluation instead of an intervention study, which was compensated by the participants’ high familiarization. Another important aspect is the data sampling until 30 min post-exercise, while the optimal data collection should also be performed 24 h post exercise. The analyzed hormones have a liner response and the efflux into the bloodstream could be maintained for 24 h post exercise, which requires further research.

## 5. Conclusions

A fast eccentric tempo seems to be more useful in eliciting a significant post-exercise hormonal stimulus compared to medium eccentric movement tempo. Hockey players respond more intensively to a higher number of repetitions than to longer time under tension in non-specific resistance exercises. Therefore, squats and the bench press exercises with a fast eccentric tempo should be used in ice-hockey during periods of non-specific resistance training programs. Considering that squats and/or bench press exercises are routinely used as interventions in numerous studies, the presented results and practical implications can be transferred to other sports disciplines.

## Figures and Tables

**Figure 1 ijerph-18-07694-f001:**
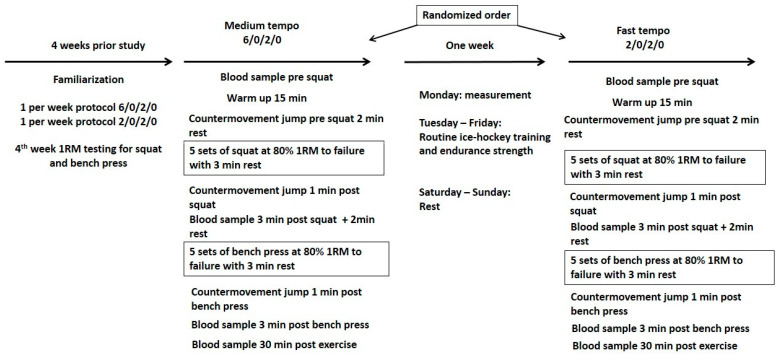
Flow chart of the experiment.

**Figure 2 ijerph-18-07694-f002:**
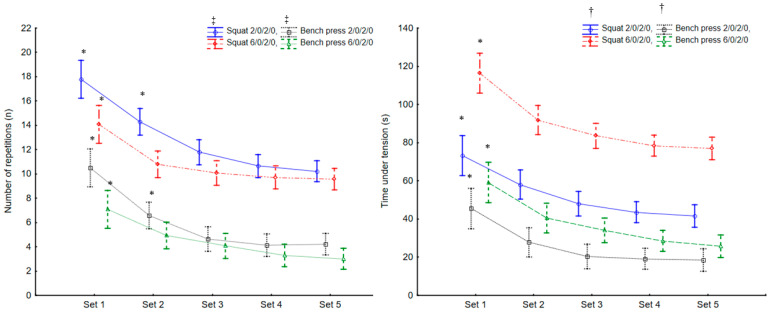
The number of repetitions and time under tension in all sets during the fast (2/0/2/0) and medium (6/0/2/0) squat and bench press protocol. * significantly higher (*p* < 0.001) than following set, ‡ significantly higher than 6/0/2/0 tempo protocol, † significantly lower (*p* < 0.001) than 6/0/2/0 TEMPO protocol. Values are expressed in mean and standard deviations.

**Figure 3 ijerph-18-07694-f003:**
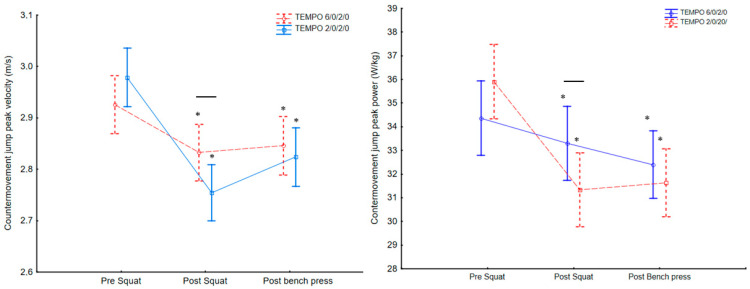
Fatigue in the countermovement jump, expressed as changes in velocity and relative peak power output during fast (2/0/2/0) and medium movement tempo (6/0/2/0) of the squat and bench press exercise protocols. * significantly lower (*p* < 0.05) than pre-squat values, dash-line shows significant differences (*p* < 0.05) between 6/0/2/0 and 2/0/2/0 protocols. Values are expressed in mean and standard deviations.

**Figure 4 ijerph-18-07694-f004:**
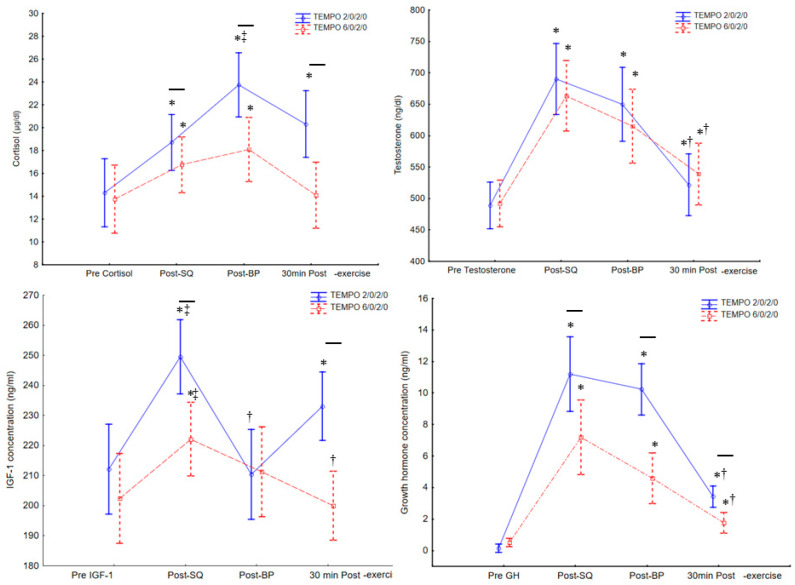
Cortisol, growth hormone, testosterone, and IGF-1 concentrations during the squat and bench press exercises with fast (2/0/2/0) and medium (6/0/2/0) movement tempo. * significantly higher (*p* < 0.05) than pre-squat values, dash-line shows significant differences (*p* < 0.05) between 6/0/2/0 and 2/0/2/0 protocols. ‡ significantly higher (*p* < 0.05) than all other values at the same protocol. † significantly lower (*p* < 0.05) than post-squat and post-bench press values, Values are expressed in mean and standard deviations. GH = growth hormone, IGF-1 = insulin-like growth factor 1.

**Figure 5 ijerph-18-07694-f005:**
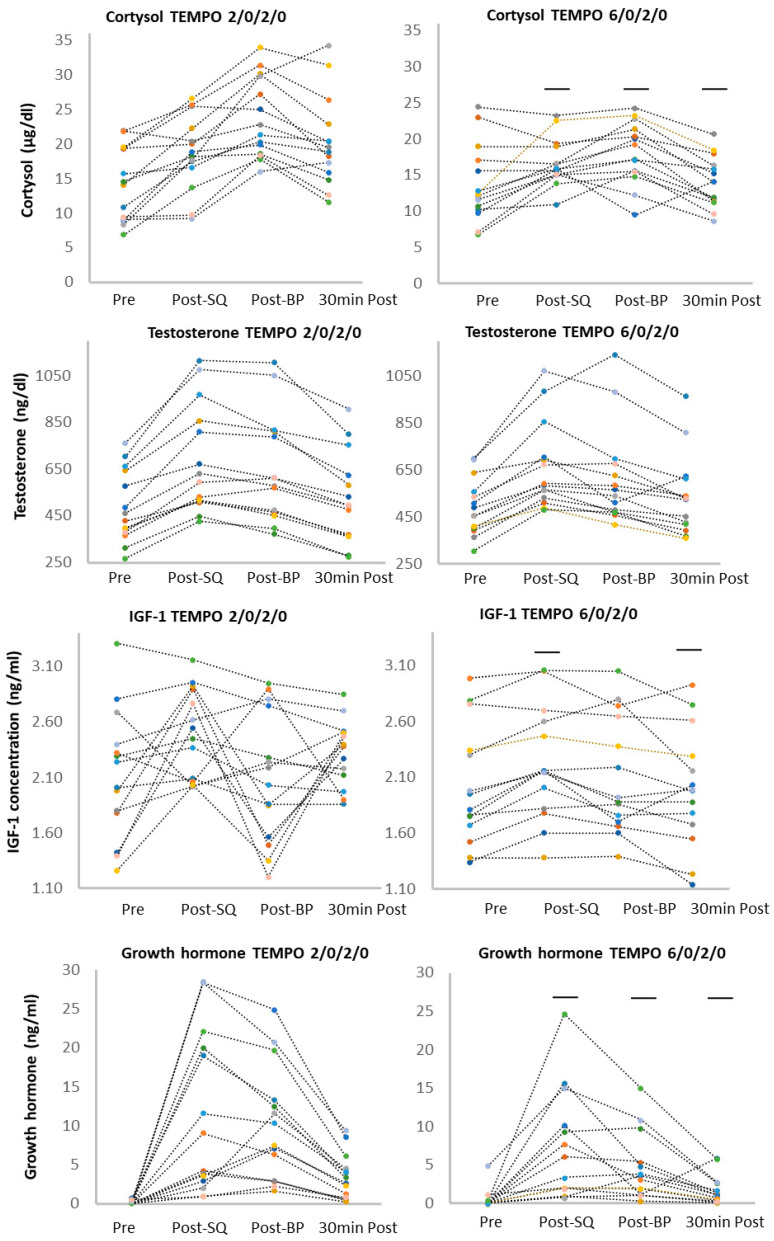
Physiological responses of cortisol, growth hormone, testosterone, and IGF-1 during the squat and bench press exercise protocols for fast (2/0/2/0) and medium (6/0/2/0) movement tempo. GH = growth hormone, IGF-1 = insulin-like growth factor. The dash line indicates the significant difference between TEMPO protocols, lower values for 6/0/2/0.

**Table 1 ijerph-18-07694-t001:** Endocrine and neuromuscular responses to fast (2/0/2/0) and medium tempo (6/0/2/0) exercise protocols.

	Medium TEMPO				Fast TEMPO			
	Pre-Squat	Post-Squat	Post-Bench Press	Post-Session 30 min	Pre-Squat	Post-Squat	Post-Bench Press	Post-Session 30 min
CMJ Vmax (m/s)	2.93 ± 0.05; 2.81–3.04	2.83 ± 0.04; 2.72–2.94	2.85 ± 0.06; 2.73–2.96	--	2.98 ± 0.06; 2.86–3.09	2.75 ± 0.06; 2.64–2.87	2.82 ± 0.06; 2.71–2.94	--
CMJ Pmax (W/kg)	34.36 ± 1.43; 31.16–37.56	33.31 ± 1.27; 30.12–36.49	32.40 ± 1.55; 29.49–35.31	--	35.90 ± 1.69; 32.70–39.10	31.33 ± 1.81; 28.15–34.52	31.63 ± 1.28; 28.73–34.54	--
Cortisol (µg/dL)	13.74 ± 1.45; 10.59–16.87	16.76 ± 0.88; 14.84–18.66	18.08 ± 1.15; 15.59–20.57	14.08 ± 0.94; 12.04–16.13	14.30 ± 1.45 11.33–17.27	18.72 ± 1.43; 16.26–21.17	23.76 ± 1.55; 20.95–26.57	20.32 ± 1.75; 17.42–23.22
GH (ng/mL)	0.52 ± 0.35; −0.24–1.27	7.20 ± 1.92; 3.04–11.34	4.59 ± 1.16; 2.06–7.11	1.76 ± 0.52; 0.65–2.88	0.15 ± 0.05; 0.04–0.27	11.21 ± 2.75; 5.26–17.15	10.23 ± 1.98; 5.96–14.50	3.43 ± 0.78; 1.73–5.12
IGF-1 (ng/mL)	202.43 ± 14.25; 171.79–233.07	222.07 ± 13.54; 192.81–251.32	211.29 ± 13.96; 181.11–241.46	200.00 ± 14.08; 169.57–230.42	212.14 ± 15.53; 178.58–245.70	249.50 ± 10.85; 226.05–272.95	210.43 ± 15.74; 176.42–244.43	233.07 ± 7.91; 215.99–250.15
Testosterone (ng/dL)	492.10 ± 32.24; 422.42–561.76	663.43 ± 49.45; 556.58–770.28	615.01 ± 55.70; 494.68–735.34	539.05 ± 45.80; 440.08–638.01	488.90 ± 41.36; 399.69–578.19	690.27 ± 62.50; 555.23–825.30	649.91 ± 62.14; 515.65–784.17	521.81 ± 51.85; 409.80–633.82

Values are expressed as mean ± SE; 95% CI. GH = growth hormone, IGF-1 = insulin like growth factor, CMJ = countermovement jump test, Vmax = maximum concentric velocity, Pmax = maximum concentric power.

**Table 2 ijerph-18-07694-t002:** Cohen *d* effect sizes for endocrine and neuromuscular changes after fast (2/0/2/0) and medium tempo (6/0/2/0) exercise protocols.

Variable	Protocol TEMPO	Pre-Exercise Post-Squat	Post-Squat Post-Bench Press	Post-Squat Post-30 min	Pre-Exercise Post-Bench Press	Pre-Exercise Post 30 min	Post-Bench Press Post-30 min	Between TEMPO Difference Post-Squa t Post-Bench Press Post-30 min		
CMJ Vmax (m/s)	fast	0.9	0.3	-	0.6	-	-	0.4	0.1	-
	medium	0.5	0.1	-	0.3	-	-			
CMJ Pmax (W/kg)	fast	0.7	0.1	-	0.7	-	-	0.3	0.2	-
	medium	0.2	0.2	-	0.3	-	-			
Cortisol (µg/dL)	fast	0.7	0.9	0.4	1.6	1	0.3	0.5	1.2	1.2
	medium	0.7	0.3	0.8	0.9	0.1	2.9			
GH (ng/mL)	fast	1.5	0.1	1.0	2.2	1.6	1.2	0.5	0.9	0.7
	medium	1.3	0.5	1.0	1.3	0.7	0.8			
IGF-1 (ng/mL)	fast	0.8	0.8	0.5	0.0	0.5	0.5	0.6	0.0	0.8
	medium	0.4	0.2	0.4	0.2	0.00	0.2			
Testosterone (ng/dL)	fast	1.0	0.2	0.8	0.8	0.2	0.6	0.1	0.2	0.1
	medium	1.1	0.2	0.7	0.7	0.3	0.4			

GH = growth hormone, IGF-1 = insulin like growth factor, Vmax = maximum concentric velocity, Pmax = maximum concentric power.

**Table 3 ijerph-18-07694-t003:** Endocrine and neuromuscular changes after fast (2/0/2/0) and medium tempo of resistance exercise protocols (6/0/2/0).

Variable	Protocol TEMPO	Pre-Exercise Post-Squat	Post-Squat Post-Bench Press	Post-Squat Post-30 min	Pre-Exercise Post-Bench Press	Pre-Exercise Post 30 min	Post-Bench Press Post-30 min
CMJ Vmax (m/s)	fast	−0.22 ± 0.15 −0.30–−0.14	0.07 ± 0.16 −0.01–0.14	-	−0.16 ± 0.18 −0.20–0.01	-	-
	medium	−0.09 ± 0.11 −0.15–0.04	0.02 ± 0.14 −0.08–0.06	-	−0.08 ± 0.12 −0.14–0.02	-	-
CMJ Pmax (W/kg)	fast	−4.57 ± 4.90 −7.09–−2.05	0.3 ± 4.09 −1.80–2.40	-	−4.27 ± 4.81 −6.75–−1.79	-	-
	medium	−1.05 ± 3.10 −2.64–0.54	−0.91 ± 3.49 −2.70–0.88	-	−1.96 ± 3.58 −3.80–0.11	-	-
Cortisol (µg/dL)	fast	4.42 ± 3.79 2.23–6.61	5.04 ± 3.57 2.98–7.10	3.44 ± 3.49 1.42–5.46	9,46 ± 5.25 6.43–12.49	6.02 ± 7.02 1.97–10.07	−3.44 ± 3.49 −5.46–−1.42
	medium	3.02 ± 3.97 0.73–5.32	1.32 ± 3.11 −0.47–3.12	−4.00 ± 0.07 −5.77–−2.22	4.34 ± 4.22 1.91–6.78	0.34 ± 4.08 −2.01–2.07	−4.00 ± 3.07 −5.78–−2.22
GH (ng/mL)	fast	11.05 ± 10.27 5.12–19.98	−0.97 ± 4.75 −3.71–1.77	−7.77 ± 7.95 −12.37–−3.18	10.08 ± 7.38 5.82–14.35	3.27 ± 2.92 1.58–4.96	−6.80 ± 4.63 −9.49–−4.13
	medium	6.67 ± 6.8 2.70–10.64	−2.60 ± 4.34 −5.11–−0.10	−5.43 ± 5.90 −8.84–−2.03	4.07 ± 4.00 1.76–6.38	1.25 ± 2.16 −0.01–2.50	−2.82 ± 3.56 −4.88–−0.76
IGF-1 (ng/mL)	fast	37.36 ± 60 2.69–72.02	−39.07 ± 68 −78–0.24	−16 ± 33.64 −35.86–2.99	1.71 ± 30 −15.97–19.39	20.92 ± 60 −14.15–56.01	22.64 ± 62 −13.26–58.55
	medium	19.64 ± 14.18 11.45–27.83	−10.78 ± 17.58 −20.93–0.63	−22.07 ± 11.51 −28.71–15.42	8.85 ± 19.02 2.23–19.95	−2.42 ± 12.08 −9.40–4.54	−11.28 ± 24.92 −25.67–3.1
Testosterone (ng/dL)	fast	201.37 ± 100 143–259	−40.36 ± 44.61 −66–−15	−168.46 ± 66 −206–−130	161.01 ± 112 95–226	32.91 ± 78 −13–78	−128.10 ± 67 −167–−90
	medium	171.32 ± 92 117–224	−48 ± 82 −95–−1	−124 ± 72 −166–−82	122 ± 120 53–192	46 ± 93 −7–100	−75 ± 72 −117–−34

Values are expressed as mean difference ± SD and 95% confidence limit–. GH = growth hormone, IGF-1 = insulin-like growth factor, CMJ = countermovement jump test, Vmax = maximum concentric velocity, Pmax = maximum concentric power.

## Data Availability

The dataset used and/or analyzed during the current study is available from the corresponding author in response to a reasonable request. Due to patient’s data, privacy data are not made available publicly.

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
