# Peer review of "Fast Eccentric Movement Tempo Elicits Higher Physiological Responses than Medium Eccentric Tempo in Ice-Hockey Players"

_ijerph, 2021, doi:10.3390/ijerph18147694_

Round 1

Reviewer 1 Report

The purpose of the research is interesting.

However, several issues and questions need to be addressed before considering the paper for publication.

Please read and properly address to all comments made in the manuscript attached.

Author Response

Thank you very much for your time and suggestions. We have addressed your specific changes in the manuscript in yellow text. We believe the changes have improved the quality of the paper, so, thank you very much.

All answers are included in the attachment file.

Reviewer 2 Report

DEAR AUTHORS,

Manuscript entitled Fast Eccentric Movement Tempo Elicits Higher Physiological Responses than Slow Eccentric Tempo in Ice-hockey Players

GENERAL COMMENTS

Clarity of content and adequacy to scientific language was demonstrated throughout the manuscript; however, some adjustments are needed.

In order to facilitate the readers' understanding, I suggest the following changes:

#Abstract

I suggest that the authors of the not affirm the results of the present study.

Regular eccentric tempo training should be preferred in ice-hockey’s resistance training programs.

I suggest the authors add

Future studies are still needed to confirm our findings.

#Introduction

I suggest that autotes leave clearer the following

In this manner, training loads should have a excessive responsein cortisol, while a high response in growth hormone, testosterone, and insulin-like growth factor.

#Results

I suggest that the authors add a table or figure only with the effect size indicators for each variable and protocol respectively.

# Conclusions

I suggest the authors remove the following:

Therefore, squats and the bench press exercises with a fast eccentric time should be used in ice-hockey during periods of non-specific resistance training.

I suggest the authors add:

Randomized studies are needed, using the same protocol of the present study while controlling the main variables related to internal and external validity.

Author Response

DEAR AUTHORS,
Manuscript entitled Fast Eccentric Movement Tempo Elicits Higher Physiological Responses than Slow Eccentric Tempo in Ice-hockey Players
GENERAL COMMENTS
Clarity of content and adequacy to scientific language was demonstrated throughout the manuscript; however, some adjustments are needed.

Reply –  Thank you very much for your time and suggestions. We have addressed your specific changes in the manuscript in yellow text. We believe the changes have improved the quality of the paper, so, thank you very much. 

In order to facilitate the readers' understanding, I suggest the following changes
#Abstract

I suggest that the authors of the not affirm the results of the present study.
Reply – change has been made. 

Regular eccentric tempo training should be preferred in ice-hockey’s resistance training programs.
I suggest the authors add
Future studies are still needed to confirm our findings.
Reply – this sentence was rewritten 
#Introduction
I suggest that autotes leave clearer the following
In this manner, training loads should have a excessive response in cortisol, while a high response in growth hormone, testosterone, and insulin-like growth factor.
Reply – change has been made  

#Results

I suggest that the authors add a table or figure only with the effect size indicators for each variable and protocol respectively.
Reply: We have added the table with the Cohen d effect size for each difference in variable and protocols. To apply this request, we have added the Cohen d calculation to statistical section.  

# Conclusions

I suggest the authors remove the following:
Therefore, squats and the bench press exercises with a fast eccentric time should be used in ice-hockey during periods of non-specific resistance training.
Reply - as suggested, this sentence has been removed 

I suggest the authors add:
Randomized studies are needed, using the same protocol of the present study while controlling the main variables related to internal and external validity.
Reply - thank you for this suggestion. Such a sentence has been added

Reviewer 3 Report

The authors compared the acute hormonal response of fast (2/0/2/0) and slow eccentric (6/0/2/0) movement tempos during 5 sets of barbell squat and barbell bench press at 80%1RM to failure in a crossover design in elite ice hockey players.

The topic is interesting particularly for experts or athletes in ice hockey.  However, I am not sure if this topic is interesting for the reader of the journal. The manuscript has major flaws in reporting and discussion the variables (see below). It remains completely unclear why the authors selected 2/0/2/0 and 6/0/2/0 movement tempo. The arguments are difficult to follow and mostly unrelated.

In addition, as a limitation of the study, the hormonal response was only examined up to 30 after exercise, and considering that some of these hormones have a liner response and the influence could continue later up to 24h, raises the question of the practical value of the study.

Title

This study is mainly about hormonal responses rather than physiological responses. Perhaps physiological responses is not a good topic for this study.

The author affiliation number 5 is not defined.

Abstract and Keywords

Line 15, all abbreviations must be defined.

Conclusion, is muscle damage measured in this study?  Conclusions should also refer to hormonal response as the result of this study.

Introduction

One of the vague parts of this study is the introduction. The introduction is very superficial and does not give a proper rationale for the study.

As the author correctly mentioned, hormonal response as well as tempo movement to strength training (RT) is very well studied. The novelty of this study is that there is no corresponding study in hockey players. Here, the authors should explain in detail a) why they believed ice hockey players might have a different response, b) why they selected specifically 2s and 6s eccentric movement from the range of very fast to very slow (0-10s).

Instead, the authors focused on the advantage of RT for conditioning the hockey player or the general property of hockey movement.

E.g., in the 2nd paragraph "During the ice-hockey season, the resistance training should maintain or increase non-specific conditioning levels, improve specific fitnessw, and ensure the physiological response for proper recovery "

Maintaining levels, improving fitness, and proper recovery are the principles of almost all active sports, not only during the completion phase, but also throughout conditioning.

Furthermore, while these terms (during- or off ice hockey conditioning… ) are familiar to training specialists, they are not as meaningful for the normal readers of this journal and do not have much to do with the topic of the study.

Here I suggest the authors to mention previous studies in detail and focus on the physiological profile of the hockey player.

Methods

The crossover design should be explained in more detail.

The flow chart of the study is not readable.

The study was conducted with one-week intervals. There is information on supplement control, but there is no information on the routine training of the participants. How did the authors control the routine training of the participants (to avoid training interruption) considering they were professional players? The interval week was passive or include some exercise?

More details on when the study was conducted. (In season, off season,...).

Results

Figures and tables do not include p-values or significance points and can be modified with more useful information for readers for each time point.

Reporting only the amount of changes for follow up measurement in table is more informative.

Were the baseline values tested for possible significant differences? If so, more information needs to be added to the text

Although individual charts are useful, according to the topic of the study, the charts could be more relevant when pointed the between group’s changes not individual changes.

Discussion

Also, the discussion is not clearly related to the result of the study. The authors should avoid speculation and rewrite this part to explain their result with relevant publications in the literature.

The fact that well-trained ice hockey players are well adapted to isometric and controlled eccentric loading is also could be true for many other sports as well reference 13. E.g., reference number 13 was also performed in bodybuilders with at least 5 years of training experience who used it for eccentric adaptation during squats or bench press.

Squats or bench press as the intervention of this study performed routinely in many other active sports and there specific adaptation to these exercise already expected.

In addition, in the concept of hormones response, the increase in cortisol more intensively in 2/0/2/0 could be considered as a conclusion against previously adaptation to fast movement in hockey players.

Reference number 8 is also a mini-review. The author must refer to the original articles with more details and comparisons.

Overall, it is not clear in the discussion why hockey players react differently and what the possible underlying factor behind

Author Response

The authors compared the acute hormonal response of fast (2/0/2/0) and slow eccentric (6/0/2/0) movement tempos during 5 sets of barbell squat and barbell bench press at 80%1RM to failure in a crossover design in elite ice hockey players.

The topic is interesting particularly for experts or athletes in ice hockey.  However, I am not sure if this topic is interesting for the reader of the journal. The manuscript has major flaws in reporting and discussion the variables (see below). It remains completely unclear why the authors selected 2/0/2/0 and 6/0/2/0 movement tempo. The arguments are difficult to follow and mostly unrelated.

Reply - Thank you for your comment. We have addressed your specific changes in the manuscript in yellow text. We believe the changes have improved the quality of the paper, so, thank you very much. Nevertheless, we acknowledge that with the large number of changes made in this draft, there may be other areas that would now require your comments and our amendments.

In addition, as a limitation of the study, the hormonal response was only examined up to 30 after exercise, and considering that some of these hormones have a liner response and the influence could continue later up to 24h, raises the question of the practical value of the study.

Reply -  I agree with the reviewer's opinion and accordingly, we have added a note on this limitation of the study

Title

This study is mainly about hormonal responses rather than physiological responses. Perhaps physiological responses is not a good topic for this study.

Reply – The results of the research concern both changes in the concentration of hormones as well as the level of power output, which is indirectly an indicator of fatigue. Therefore, in order to include both hormonal and power output changes, the term physiological responses were used. In line with this, we decided to leave the topic unchanged

The author affiliation number 5 is not defined.

Reply - this error has been corrected

Abstract and Keywords

Line 15, all abbreviations must be defined.

Reply – done

Conclusion, is muscle damage measured in this study?  Conclusions should also refer to hormonal response as the result of this study.

Reply - as suggested, we made changes in the conclusion

Introduction

One of the vague parts of this study is the introduction. The introduction is very superficial and does not give a proper rationale for the study.

Reply – the rationale of the study was add.

As the author correctly mentioned, hormonal response as well as tempo movement to strength training (RT) is very well studied. The novelty of this study is that there is no corresponding study in hockey players. Here, the authors should explain in detail a) why they believed ice hockey players might have a different response, b) why they selected specifically 2s and 6s eccentric movement from the range of very fast to very slow (0-10s).

Instead, the authors focused on the advantage of RT for conditioning the hockey player or the general property of hockey movement.

Reply –thank you very much for this valuable attention. Yes not only there are no studies about the impact of movement tempo in hockey players, but most of all there is a lack of this such research in the group of people advanced in resistance training and also habitual to training with different movement tempos. We have added an additional justification for undertaking the research as well as an explanation of why these tempos were used.

E.g., in the 2nd paragraph "During the ice-hockey season, the resistance training should maintain or increase non-specific conditioning levels, improve specific fitnessw, and ensure the physiological response for proper recovery "

Maintaining levels, improving fitness, and proper recovery are the principles of almost all active sports, not only during the completion phase, but also throughout conditioning.

Furthermore, while these terms (during- or off ice hockey conditioning… ) are familiar to training specialists, they are not as meaningful for the normal readers of this journal and do not have much to do with the topic of the study.

Here I suggest the authors to mention previous studies in detail and focus on the physiological profile of the hockey player.

Reply – we added the information about the detail physiological profile of the hockey player.

Methods

The crossover design should be explained in more detail.

The flow chart of the study is not readable.

Reply – the flow chart was corrected and we believe that now the crossover design is clear

The study was conducted with one-week intervals. There is information on supplement control, but there is no information on the routine training of the participants. How did the authors control the routine training of the participants (to avoid training interruption) considering they were professional players? The interval week was passive or include some exercise?

More details on when the study was conducted. (In season, off season,...).

Reply - relevant information has been added

Results

Figures and tables do not include p-values or significance points and can be modified with more useful information for readers for each time point.

Reply: Sorry for this inaccuracy, the figures now include the significant marks, which are described in the figure caption. There is also one more table with pair effect sizes to see the effect of each condition.

Reporting only the amount of changes for follow up measurement in table is more informative.

Reply: We have added Table 3 which includes the main values changes in both protocols.

Were the baseline values tested for possible significant differences? If so, more information needs to be added to the text.

Reply: Yes, they were tested according to the statistical section and show no differences. This we now stated in the first paragraph of the result after the normality report.

Although individual charts are useful, according to the topic of the study, the charts could be more relevant when pointed the between group’s changes not individual changes.

Reply: We agree, therefore we have added the between-group (condition) differences in Figure 5. However, the between-group differences are already in figures 3 and 4.

Discussion

Also, the discussion is not clearly related to the result of the study. The authors should avoid speculation and rewrite this part to explain their result with relevant publications in the literature.

Reply – the discussion has been improved and we believe that it will now meet the expectations of the reviewers

The fact that well-trained ice hockey players are well adapted to isometric and controlled eccentric loading is also could be true for many other sports as well reference 13. E.g., reference number 13 was also performed in bodybuilders with at least 5 years of training experience who used it for eccentric adaptation during squats or bench press.

Squats or bench press as the intervention of this study performed routinely in many other active sports and there specific adaptation to these exercise already expected.

Reply – yes, we agree that these exercises are fundamental to many sports disciplines, therefore the results of this research may also be useful for other sports. The relevant information has been added to the conclusion

In addition, in the concept of hormones response, the increase in cortisol more intensively in 2/0/2/0 could be considered as a conclusion against previously adaptation to fast movement in hockey players.

Reply – we agree that an increase in cortisol more intensively in 2/0/2/0 could be considered as a conclusion against previously adaptation to fast movement, however, this subject was familiar to regular resistance training with a different tempo of movement also slow.

Reference number 8 is also a mini-review. The author must refer to the original articles with more details and comparisons.

Reply – we added original refry

Overall, it is not clear in the discussion why hockey players react differently and what the possible underlying factor behind

Reply - thanks for this comment, a proper explanation has been added

Round 2

Reviewer 3 Report

The authors have addressed most of the issues, the revised manuscript has greatly improved, but there are still some issues that need to be addressed, and some errors need to be corrected.

  • Because the topic has already changed (Fast vs. Medium). Throughout the text, these changes should be implied. It is a bit confusing when the authors still talking about slow tempo or (slower tempo without refereeing the time) in the manuscript. For example:

  • in the abstract (conclusion).
  • introduction (lines 71, 72, 74, 129)
  • and overall Discussion (slower tempo…)

It is not clear when the authors talking about slow tempos whether they mean tempo above 10 S or a tempo lower than the fast tempos. Also, in discussion, the term slower tempo is not transparent. For example, Medium tempo is a slower tempo compared to fast tempo.  The manuscript can be more accurately written when referring to the tempo.

  • The flow chart of interval week should be updated. It seems the participants had only 2 days rest. If there is extra information about training plan in others days can be useful for readers.

  • Tables are inconsistence and hard to understand. We can see Table 1 is mean ± SE, Table 3 is mean ± SD and  mean ± CI.  Here, similar to the charts, the authors can use same abbreviation for tables.  

It is better to just keep all tables similar and report data only based on one of variability parameters (SE, SD or CI) and same format for all of the tables and charts.

In addition, the data in tables need to be recheck for possible errors. F.e table 3 pre IGF1 (2.69 – 72-02) and cortisol 9,.46 ± 5.25.

The authors already defined the Medium and fast tempo in the introduction but still in charts and tables sometime, they refer the tempo as time (2/0/2/0) sometimes both together.

Here it better to keep everything simple as Medium vs fast according to the manuscript title.

Author Response

The authors have addressed most of the issues, the revised manuscript has greatly improved, but there are still some issues that need to be addressed, and some errors need to be corrected.

Answer: We would like to thank reviewer for his constructive comments and suggestions that helped us to improve our manuscript. We greatly appreciate the input and we hope that the changes made according to the reviewer suggestions addressed all the raised issues and improved the manuscript

Because the topic has already changed (Fast vs. Medium). Throughout the text, these changes should be implied. It is a bit confusing when the authors still talking about slow tempo or (slower tempo without refereeing the time) in the manuscript. For example:

in the abstract (conclusion).

introduction (lines 71, 72, 74, 129)

and overall Discussion (slower tempo…)

It is not clear when the authors talking about slow tempos whether they mean tempo above 10 S or a tempo lower than the fast tempos. Also, in discussion, the term slower tempo is not transparent. For example, Medium tempo is a slower tempo compared to fast tempo.  The manuscript can be more accurately written when referring to the tempo.

Answer: I agree with the reviewer, so we made corrections in the area of the entire article

The flow chart of interval week should be updated. It seems the participants had only 2 days rest. If there is extra information about training plan in others days can be useful for readers.

Answer: We agree, and we have added the whole week contend to the flow chart in Figure 1.  

Tables are inconsistence and hard to understand. We can see Table 1 is mean ± SE, Table 3 is mean ± SD and  mean ± CI.  Here, similar to the charts, the authors can use same abbreviation for tables. 

It is better to just keep all tables similar and report data only based on one of variability parameters (SE, SD or CI) and same format for all of the tables and charts.

Answer: We believe that our approach of reporting the values in tables and figures make sense and is appropriate. The table 1 represent the mean ± SE because the same mean values are expressed in figures 3 and 4 with graphic expression of SD. Thus we don’t want to double the same distribution data and we are presenting both major data distribution values.  In the table 3 are means ± SD and confidence limits range, which we believe are best to describe data distribution for delta values (there are no figures for this data).  Although we understand that there is some small inconsistency, in this form the data meaning is clearly presented in the tables and figures.

In addition, the data in tables need to be recheck for possible errors. F.e table 3 pre IGF1 (2.69 – 72-02) and cortisol 9,.46 ± 5.25.

Answer: Thank you for this check. We have corrected several typos in the tables.

The authors already defined the Medium and fast tempo in the introduction but still in charts and tables sometime, they refer the tempo as time (2/0/2/0) sometimes both together.

Answer: We are reporting in tables and figures always both ways, because Medium/fast is definition for the text. However, each figure and table should be possible to fully interpret alone. Therefore, such information, what is exact value of fast and medium is need it.

On the other hand, we agree that some numerical tempo description should be amended in the tables. Therefore we have revised the tables for such simplification.

Here it better to keep everything simple as Medium vs fast according to the manuscript title.

Answer: We understand that there is a conflict of simple meaning and keeping the figure and table to be interpretable alone with most significant facts. Therefore we prefer to keep both descriptions (fast/medium and tempo time) in figures and tables.